## [Reviewer comments · Life Science Alliance]

Tissue-specific I-Smad mechanisms revealed by structure-function analysis in *Drosophila*

Ania Simoncek, Steven Sviridoff, Joshua Hays, Noah Graichen, and Mikolaj Sulkowski
DOI: <https://doi.org/10.26508/lsa.202503445>

Corresponding author(s): Mikolaj Sulkowski, Southern Connecticut State University

Review Timeline:	Submission Date:	2025-07-07
	Editorial Decision:	2025-08-20
	Revision Received:	2025-11-26
	Editorial Decision:	2025-12-22
	Revision Received:	2026-02-04
	Accepted:	2026-02-08

Scientific Editor: Tim Fessenden

Transaction Report:

August 20, 2025

Re: Life Science Alliance manuscript #LSA-2025-03445-T

Mikolaj J Sulkowski
Southern Connecticut State University

Dear Dr. Sulkowski,

Thank you for submitting your manuscript entitled "Tissue-specific I-Smad mechanisms revealed by structure-function analysis in *Drosophila*" to Life Science Alliance. The manuscript was assessed by expert reviewers, whose comments are appended to this letter.

As you will see, reviewers appreciated the new insights into tissue-specific BMP signaling regulation by inhibitory SMADs. While reviewers were somewhat mixed in their overall enthusiasm, some consistent points emerged across the reviews and must be addressed in a revision. As noted in cross-commenting by Reviewers 2 and 3, a revised manuscript should clearly acknowledge potential artifacts of protein overexpression and include some justification for the use of ap-Gal4. In addition, the observations supporting signaling in wing development should be strengthened according to the suggestions by Reviewer 3, who also sought clarification of sample size, sex differences, and cut-off values for quantifications. Finally, Reviewer 2 made several important suggestions to improve the clarity of the text and figures. While a CRISPR-mediated knock-in approach would offer a valuable alternative to gene overexpression, we concur that this is not required in a revision.

Thank you for this interesting contribution to Life Science Alliance. We are looking forward to receiving your revised manuscript.

Sincerely,

B. MANUSCRIPT ORGANIZATION AND FORMATTING:

Reviewer #1 (Comments to the Authors (Required)):

This study explores the tissue-specific regulatory mechanisms of inhibitory Smads (I-Smads), *Drosophila* Dad and its vertebrate orthologs Smad6 and Smad7, in the context of BMP signalling. Through structure-function analyses in wing and neural tissues, the authors find a DNA-binding domain (DNABD) within Dad's MH1 domain that is essential for BMP inhibition in wing tissue but dispensable in neural tissue. The study further suggests that Dad requires an intact MH1 domain for wing development, while either MH1 or MH2 domains can independently inhibit BMP signalling in neurons. Comparative analysis of Smad6 and Smad7 supports the hypothesis of divergent, tissue-specific regulatory mechanisms.

While the manuscript presents potentially interesting findings, several aspects of the experimental design and data interpretation require substantial revision.

Below, I outline key concerns and suggestions for improvement.

- The authors rely on ectopic overexpression of I-Smads and their variants. Given that I-Smads function within feedback loops of BMP signalling, overexpression may not accurately reflect physiological roles and could lead to misleading conclusions. To address this, I recommend employing CRISPR/Cas9-mediated knock-in strategies to express I-Smads under endogenous regulatory control. Please refer to Akiyama et al., *eLife* 2018 (DOI: 10.7554/eLife.352581) for relevant methodology.
- The rationale for using *ap-Gal4* to assess wing phenotypes is unclear. The *ap-Gal4* drives expression in the dorsal compartment, which may not be optimal for evaluating wing-pouch-specific effects. Drivers such as *nubbin-Gal4* or *rotund-Gal4* would be more appropriate. The authors should justify their choice or consider alternative drivers.
- Adult wing phenotypes alone are not sufficient to evaluate BMP signalling during wing development. Direct readouts, such as anti-phospho-Smad antibody staining in larval wing imaginal discs and pupal wings, are necessary to validate BMP signalling. Without such data, comparisons between wing and neural tissues lack mechanistic support.
- The study compares the functional effects of Dad, human Smad6, Smad7, and their variants based on ectopic expression. However, protein levels in target tissues are not assessed. Quantitative evaluation using antibody staining or fluorescently tagged constructs is required to confirm expression and ensure comparability across constructs.

Reviewer #2 (Comments to the Authors (Required)):

Summary of the main findings

In this manuscript Simoncek and colleagues investigate the tissue-specific mechanisms of I-Smad function in *Drosophila melanogaster*, focusing on the inhibitory Smad protein Dad and its vertebrate orthologs Smad6 and Smad7 function in wing and neural tissues. Through a series of structure-function analyses, they identify a 24-amino acid putative DNA-binding motif (Δ DNABD) in MH1 domain of Dad essential for inhibition in the wing but not in neural tissue. The Dad inhibitory function specifically requires the MH1 domain in the wing tissue, while in the motor neurons, either the MH1 or MH2 domain is sufficient to inhibit BMP signaling. This study uses *Drosophila* genetics and functional assays in wing and neural tissues to investigate how Dad, Smad6, and Smad7 inhibit the BMP pathway. The use of domain deletions, human ortholog expression, and AlphaFold modeling suggests that these inhibitors operate through distinct, tissue-dependent modes involving either a direct transcriptional repression or interference with receptor/Smad signaling.

Overall, this is a well written manuscript, and the experiments are clearly presented and technically rigorous and provide new insights into context-specific regulation of BMP signaling by I-Smads.

Major comments

1. Authors have primarily used the transgenic overexpression approach to unravel the distinct inhibitory functions of Dad in wing and neural tissues. While this approach works for them, overexpression can lead to non-physiological interactions or artifacts. This limitation should be explicitly acknowledged and discussed in the discussion section.
2. Throughout the manuscript, multiple figures lack corresponding mention in the main text.
3. The schematic in Figure 1A is not informative regarding the exact deletion used to generate Δ DNABD and Δ Trimer. Authors should consider including a schematic of the conserved regions and the full sequence alignment of Dad, Smad6 and Smad7 in the supplementary. Also highlight the key residues mutated or deleted to generate various constructs used in this study.
4. For data presented in Figure 1B, representative images should be provided here or in the supplementary.
5. The Brp punctae in Figure 3A is not obvious in this image. Better representative image should be provided here.
6. The representative image and quantification of the pMad and TwitGFP panel for Δ MH2 in Figure 4A and B do not match. Authors should provide a better representative image which matches the quantification.
7. Line 206-207 - " Δ MH2 reduced Twit-GFP to 0.734 ± 0.083 , but this reduction was not statistically significant". However Figure 4B shows a single * significance.
8. The finding that Dad-C556A, a palmitoylation deficient version of Dad, is fully functional in wing and neuronal tissues is an interesting one. The observation does suggest that C556 mediated palmitoylation is dispensable in this context. To strengthen this conclusion, the authors should test if neuronal overexpression of UAS-Dad-C556A will phenocopy the synaptic undergrowth when Dad is overexpressed (PMID: 27671198).

Minor Comments

1. Sample sizes (n) and detailed statistical methods should be clearly stated in each figure legend for transparency and reproducibility.
2. The labeling of the constructs should be consistent. Consider labelling Δ DNABD as Dad Δ DNABD and so forth.
3. Smad7L is not defined in the text.
4. Elav in Figure 2D/4A/5B and 6A should be labelled.
5. Data cannot be cited as "data not shown" in the article (Line 188).
6. HRP in Figure 3A should be labelled.

Referee Cross-Comments

1. I agree with the comments raised by reviewer 1 and 3. Authors should definitely justify the use of Ap-Gal4 for evaluating the effect of BMP signaling on wing development. Regarding reviewer 3's point, the authors should also discuss the limitations of the functional assays used to assess BMP signaling in both wing and neural tissues.
2. I recognize that using CRISPR/Cas9 to drive gene expression from endogenous promoters would yield physiologically relevant insights. However, I do not consider it necessary for the authors to repeat their experiments using knock-in approaches for this study, as this would require significant additional time and resources.

Reviewer #3 (Comments to the Authors (Required)):

In this manuscript, Simoncek et al investigate the relative contributions of the I-Smad MH1 and MH2 domains in two well-studied Drosophila in vivo model tissues: signaling outputs at the larval neuromuscular junction synapse (NMJ) and the development of the adult wing blade. Drosophila I-Smad Dad is compared to human(?) Smad6 and Smad 7, using a transgenic forced expression system in each tissue. Specifically, the authors focus on the relative contributions of the Dad MH1 and MH2 domains to forced expression phenotypes within each tissue. The MH1 and MH2 domains of RSmads and CoSmads are well-studied; in these Smad classes the MH1 domain contains a documented DNA binding domain, and the MH2 domain contains the Smad trimerization region as well as the RSmad C-terminal site for phosphorylation by activated BMP receptors. The function of the MH1 domain in I-Smads is poorly understood, and may be distinct between the two mammalian I-Smads, Smad6 and Smad7, as assayed in cultured cell lines.

The work presented in this manuscript focuses on in vivo strategies to examine structure-function questions about each I-Smad domain, comparing different signaling outputs in different tissues. In general, the authors report intriguing results, but limitations to the study are not mentioned, perhaps due to the limited length for this MS format. Overall, the manuscript might benefit from a tighter focus on only a subset of this combination of studies.

The assays used for signaling output are distinct between the two tissues. Three relatively-direct BMP signaling outputs are investigated in the NMJ: 1) expression of twit gene induced by "canonical" BMP signaling in larval motor neurons, assayed by GFP protein fluorescence intensity; 2) the associated BMP-induced nuclear PMad accumulation in motor neurons, assayed by intensity of nuclear antiPMad immunofluorescence, and 3) a "non-canonical" accumulation of PMad at the synapse membrane, assayed by antiPMad immunofluorescence intensity in NMJ synaptic boutons. For overall wing development, one indirect, composite signaling output is measured: the area of the adult wing. Most of the limitations are related to the wing development

assay.

--Major comments:

The NMJ assays of direct, canonical signaling output are useful to assess the contributions of the relative functions of Dad full-length, delta-MH1, and delta-MH2. In these assays the authors find that both delta-MH1 and delta-MH2 impair nuclear pMad accumulation, but that Dad-delta-MH1 has a much stronger effect on Twit-GFP expression levels than does Dad-delta-MH2. The latter result could be consistent with a direct function of the MH1 domain on gene expression, but also could support any other potential contribution of the MH1 domain. The data of Figure 5 indicate that the delta-DNA binding domain construct has a stronger effect in blocking Twit-GFP expression, than for reducing pMad accumulation. This distinction is the key observation that a putative Dad DNA binding domain has a strong contribution to Dad-mediated antagonism of BMP-induced gene expression.

While this result supports a potential independent transcriptional function for the DNA-binding domain, the authors miss an opportunity for a second test of this proposed functional output. Expression of full-length Dad impairs the "non-canonical signaling" output of pMad accumulation in the NMJ synaptic boutons, far from the nucleus. Their hypothesis of a transcriptional function for Dad MH1 predicts that the delta-MH1 construct would not impair Mad accumulation at the synapse. The authors could at least discuss this (and possibly other) additional tests, it is not essential for publication of the current work.

The wing area assay is used to compare the effects of forced expression of full-length Dad, Smad6, and Smad7, as well as the effects of a Dad MH1-deletion construct, a Dad MH2 deletion construct, a Smad7 MH2 deletion construct, a Dad mutant for the putative DNA binding domain, a Dad mutant for a putative palmitoylation domain, and a Dad mutant for a poorly discussed "Trimer interface." The authors interpret this area measurement as "growth", but wing area is a complex output of BMP signaling activity. The distinction between small overall allometric changes in growth versus a large change in wing area due to defects in wing patterning is apparent in Fig. 5D, where patterning of wing veins is collapsed at the anterior boundary by the delta-Trimer construct, but only allometric growth appears to be altered in the representative wings for the delta-MH1 or delta-MH2 constructs. (See below for a query about the sex of the smaller wings in Fig 4C and 5D.) Even so, this assay is useful for comparisons of general Dad activity during wing development, as presented in lines 105 - 137.

Unfortunately, the wing 2D area assay used has limitations for a nuanced structure-function analysis, which are not mentioned in the interpretations of the data. First, area is impacted by BMP-directed apportionment of cells to vein and intervein cell differentiation, which is a developmental output, but not a "growth" output. Second, the BMP signal activity gradient is feedback-regulated by multiple gene products acting at various levels of the BMP signal transduction pathway, so that the gradient of BMP activity output scales with growth of the developing wing primordium. The forced expression phenotypes appear dominated by the MH2-domain-mediated Dad functions, but any MH1-mediated functions could be swamped out by activity of other feedback loops and regulatory interactions with other signals that control growth (e.g. Wnt).

The use of the Ap-Gal4 gene expression driver may confound an accurate measurement of wing area in many samples, because the highest level of forced expression is limited to the dorsal portion of the developing wing. Differential growth of dorsal and ventral sides of the wing blade can lead to 3D changes in structure, such as a curled or cup-shaped wing, which confound wing area measurement from a "flat-mounted" prep. Some of the wings shown in Fig 2C (Dad, Smad6, Smad7L) have the appearance of curled/cupped/folded wing blades. Area of such wings gives an estimate of the relative activities of the different constructs, but is unlikely to provide insight into the specific intracellular functions of specific structures within the Dad protein.

Lastly, more information is needed about the "representative" wings shown for phenotypes of different constructs in Figs 4C and 5D. Are these wings from the data sets used for area quantification in Fig 1B? Are the wings shown in each panel all from the same sex? Male wings are allometrically smaller than female wings, so it is critical to only compare wings from the same sex. If these panels each compare a female control wing with a male wing for either or both overexpression conditions, they should be corrected to show all wings from the same sex.

If the allometrically reduced wing growth of some examples in Figs 4C and 5D is present in a significant proportion of wings from one sex, then it would be a significant result. Addition of qualitative analyses of wing phenotypes to distinguish between classes of developmental perturbations would require a minimal additional analysis of available data, and could be important to identify candidate target genes that are specifically perturbed by MH1-domain specific functions.

Finally, the authors control for consistent transcriptional output of the different constructs, by using the same insertion site of VK00033. However, each of the constructs may have differential effects on protein stability, resulting in different steady-state protein levels. This possibility should be raised in the discussion, because it is a significant limiting factor to interpretation of constructs that appear to have no or low activity in one or both tissues.

Overall, the manuscript presents both descriptive and quantified data that are interesting with regard to functions of different I-Smad conserved domains. Acknowledgement of the limitations of each assay, more careful documentation of the wing area assay, and better description of sample size and cut-offs for statistical significance are necessary minor revisions. If warranted

as described above, corrections to figures 4C and 5D are essential for publication.

--Minor comments:

The measurement of wing area is poorly described: it is not clear whether the authors measure wing blade +hinge (and possibly associated bits of body wall, as seen in Fig 2C Smad6 image), or just the wing blade area, for which developmental regulation is better understood.

The authors report "mean wing areas", but do not report the sample sizes.

For graphs of normalized intensity in Figs. 2E, 3B, 4B, 5C, 6B, sample sizes should be stated in Figure legends; they are difficult to assess from the dot plots.

The authors should explicitly state a cut-off for statistical significance in quantified measurements of wing area, Twit-GFP expression or PMad nuclear accumulation. In Fig. 5D, is the effect of deltaDNABD on TwitGFP expression non-significant?

Red color used to show PMad accumulation in micrographs is hard to see. Using white instead of red would make this output readily visible to the reader.

--Referee Cross-Comments--

Thoughts on Reviewer 1 Comments;

While CRISPR-based Dad gene replacement would be a superior approach compared to overexpression, such additional experiments would take at least a year if performed by early-stage graduate students.

Thoughts on Reviewer 2 Comments:

I agree with major comment 3: the specific deletions of deltaDNADB and delta-Trimer should be shown with Fig 1 diagrams that indicate the specific residues deleted.

Major comment 4 is related to my request for information on qualitatively different wing blade phenotypes. However, a single "representative" image may not be sufficient to represent a potential range of developmental outcomes.

Regarding major comments 5 and 6: Each of the markers used could be shown in black and white images, so that differences in intensity are easily detected by the human eye. Note that the "merge" label in Fig 2 appears to be incorrect.

Major comment 7 fits in with my concern about whether the authors have a designated cut-off for statistical significance.

Dear Dr. Fessenden,

Thank you for your consideration of our manuscript (#LSA-2025-03445-T) entitled “Tissue-
specific I-Smad mechanisms revealed by structure-function analysis in *Drosophila*” for
publication. We are deeply appreciative of the reviewer’s expert feedback and your guidance on
key concerns regarding choice of wing expression driver, caveats and limitations of our
experimental paradigm, and clarification over sample size, sex differences, and cut-off values for
quantifications.

In our revision, we have carefully addressed these points by performing additional experiments
with an alternative wing-specific Gal4 line, overhauling all figures and legends, providing
comprehensive statistical information, and acknowledging limitations. At the same time, we
emphasize the strengths of our study. By analyzing the effects of I-Smad structural motifs in
different tissue types, our work provides novel insight into how a single molecule can function
through alternative mechanisms in different cellular contexts.

We believe these revisions substantially improve the clarity and impact of the manuscript, and
we hope you will now find it suitable for publication in *Life Science Alliance*.

Please find our point-by-point responses to reviewer comments below. The original comments
are reproduced in regular font while our responses are shown in blue font.

Sincerely,

The Authors

**Scientific Editor's Comments to Author:**

Thank you for submitting your manuscript entitled "Tissue-specific I-Smad mechanisms revealed
by structure-function analysis in *Drosophila*" to Life Science Alliance. The manuscript was
assessed by expert reviewers, whose comments are appended to this letter.

As you will see, reviewers appreciated the new insights into tissue-specific BMP signaling
regulation by inhibitory SMADs. While reviewers were somewhat mixed in their overall
enthusiasm, some consistent points emerged across the reviews and must be addressed in a
revision. As noted in cross-commenting by Reviewers 2 and 3, a revised manuscript should
clearly acknowledge potential artifacts of protein overexpression and include some justification
for the use of ap-Gal4. In addition, the observations supporting signaling in wing development
should be strengthened according to the suggestions by Reviewer 3, who also sought clarification
of sample size, sex differences, and cut-off values for quantifications. Finally, Reviewer 2 made
several important suggestions to improve the clarity of the text and figures. While a CRISPR-
mediated knock-in approach would offer a valuable alternative to gene overexpression, we
concur that this is not required in a revision.

**Reviewer Comments to Author:**

**Reviewer 1:**

This study explores the tissue-specific regulatory mechanisms of inhibitory Smads (I-Smads),
*Drosophila* Dad and its vertebrate orthologs Smad6 and Smad7, in the context of BMP
signalling. Through structure-function analyses in wing and neural tissues, the authors find a
DNA-binding domain (DNABD) within Dad's MH1 domain that is essential for BMP inhibition
in wing tissue but dispensable in neural tissue. The study further suggests that Dad requires an
intact MH1 domain for wing development, while either MH1 or MH2 domains can
independently inhibit BMP signalling in neurons. Comparative analysis of Smad6 and Smad7
supports the hypothesis of divergent, tissue-specific regulatory mechanisms.
While the manuscript presents potentially interesting findings, several aspects of the
experimental design and data interpretation require substantial revision.
Below, I outline key concerns and suggestions for improvement.

1. The authors rely on ectopic overexpression of I-Smads and their variants. Given that I-Smads
function within feedback loops of BMP signaling, overexpression may not accurately reflect
physiological roles and could lead to misleading conclusions. To address this, I recommend
employing CRISPR/Cas9-mediated knock-in strategies to express I-Smads under endogenous
regulatory control. Please refer to Akiyama et al., eLife 2018 (DOI: 10.7554/eLife.352581) for
relevant methodology.

We appreciate the reviewer's suggestion to utilize gene editing technology to recreate our
deletions in endogenous loci to study tissue-specific functions. We agree that this approach

would alleviate many concerns over potentially artificial responses to forced expression, and we
have updated the Discussion to point to this methodology as a future direction (lines 385-394).
However, we agree with the editor and other reviewers that undertaking such an approach would
not be feasible for the current study. We plan to pursue these experiments in follow-up studies.

2. The rationale for using ap-Gal4 to assess wing phenotypes is unclear. The ap-Gal4 drives
expression in the dorsal compartment, which may not be optimal for evaluating wing-pouch-
specific effects. Drivers such as nubbin-Gal4 or rotund-Gal4 would be more appropriate. The
authors should justify their choice or consider alternative drivers.

We conducted experiments using Rn-Gal4 (Figure S2). This driver produced even more dramatic
phenotypes than Ap-Gal4. Notably, this driver produced wing venation defects when used to
express Dad^{ΔMH1}, indicating that Dad MH2-mediated mechanisms can have some effect on wing
development under stronger expression conditions. This provides important nuance and caveats
to our tissue-specific mechanisms hypothesis.

3. Adult wing phenotypes alone are not sufficient to evaluate BMP signalling during wing
development. Direct readouts, such as anti-phospho-Smad antibody staining in larval wing
imaginal discs and pupal wings, are necessary to validate BMP signalling. Without such data,
comparisons between wing and neural tissues lack mechanistic support.

We agree that direct measurement of pMad levels in larval wing imaginal discs would strengthen
our mechanistic conclusions about BMP signaling during wing development. We attempted to
perform anti-phospho-Smad antibody staining in wing discs; however, we were unable to obtain
reliable, interpretable results within the revision timeframe. We acknowledge this limitation in
the revised Discussion (lines 408-415).

We note two lines of evidence supporting our conclusions despite the absence of direct pMad
measurements in wing discs: (1) Adult wing morphology defects are well-established readouts of
disrupted BMP signaling during wing development (Inoue et al. 1998; O'Connor et al. 2006;
López-Varea et al. 2021); (2) Our new experiments with Rn-Gal4 (Figure S2) produce even
more dramatic phenotypes than Ap-Gal4, including wing venation defects characteristic of BMP
disruption.

While direct measurement of pMad in wing discs would be valuable for future studies, we
believe the current data provide sufficient evidence for tissue-specific differences in I-Smad
mechanisms to warrant publication, particularly given the novel insights into the role of the Dad
MH1 domain and putative DNA-binding region.

4. The study compares the functional effects of Dad, human Smad6, Smad7, and their variants
based on ectopic expression. However, protein levels in target tissues are not assessed.
Quantitative evaluation using antibody staining or fluorescently tagged constructs is required to
confirm expression and ensure comparability across constructs.

We appreciate this thoughtful comment. While we agree that direct quantification of protein
levels would be informative, we currently lack the technical ability to conduct this experiment
due to the absence of antibodies specific to Dad and the inability to generate animals expressing
tagged constructs within the revision timeframe. We note that other reviewers and the editor
suggested addressing this as an important caveat in the Discussion (lines 385-394), which we
have done.

Several aspects of our experimental design help alleviate this concern: (1) All constructs were
integrated at the same genomic locus (VK00033), which minimizes positional effects and
ensures comparable mRNA levels; (2) Each construct produces robust, reproducible phenotypes
in our assays. Constructs showing no activity or weak activity in one tissue type (e.g., Dad^{MH1}
in wing) produce consistent, strong phenotypes in other tissues (motor neurons), arguing against
technical failures in expression.

**Reviewer 2:**

Summary of the main findings

In this manuscript Simoncek and colleagues investigate the tissue-specific mechanisms of I-
Smad function in *Drosophila melanogaster*, focusing on the inhibitory Smad protein Dad and its
vertebrate orthologs Smad6 and Smad7 function in wing and neural tissues. Through a series of
structure-function analyses, they identify a 24-amino acid putative DNA-binding motif
(Δ DNABD) in MH1 domain of Dad essential for inhibition in the wing but not in neural tissue.
The Dad inhibitory function specifically requires the MH1 domain in the wing tissue, while in
the motor neurons, either the MH1 or MH2 domain is sufficient to inhibit BMP signaling. This
study uses *Drosophila* genetics and functional assays in wing and neural tissues to investigate
how Dad, Smad6, and Smad7 inhibit the BMP pathway. The use of domain deletions, human
ortholog expression, and AlphaFold modeling suggests that these inhibitors operate through
distinct, tissue-dependent modes involving either a direct transcriptional repression or
interference with receptor/Smad signaling.

Overall, this is a well written manuscript, and the experiments are clearly presented and
technically rigorous and provide new insights into context-specific regulation of BMP signaling
by I-Smads.

1. Authors have primarily used the transgenic overexpression approach to unravel the distinct
inhibitory functions of Dad in wing and neural tissues. While this approach works for them,
overexpression can lead to non-physiological interactions or artifacts. This limitation should be
explicitly acknowledged and discussed in the discussion section.

We thank the reviewer for this thoughtful comment. We have addressed this concern in the
Discussion (lines 385-407).

2. Throughout the manuscript, multiple figures lack corresponding mention in the main text.

We have carefully reviewed the manuscript and ensured that all figures are referenced in the text.
We appreciate the reviewer's careful reading.

3. The schematic in Figure 1A is not informative regarding the exact deletion used to generate
Δ DNABD and Δ Trimer. Authors should consider including a schematic of the conserved regions
and the full sequence alignment of Dad, Smad6 and Smad7 in the supplementary. Also highlight
the key residues mutated or deleted to generate various constructs used in this study

We created a new figure (Figure S1) showing the precise residues deleted in the Dad Δ DNABD
and Dad Δ Trimer constructs. This figure also includes alignments with human I-Smad homologs.
The high alignment score for the putative DNA binding region between Dad and Smad6
strengthens our hypothesis of a conserved mechanism involving this region of Smad6/Dad, but
not Smad7. We thank the reviewer for this suggestion.

4. For data presented in Figure 1B, representative images should be provided here or in the
supplementary.

The representative wing images in Figures 2C, 4C, and 5D are from wings quantified in Figure
1B. We have clarified this in multiple locations throughout the manuscript (figure legends and
Methods section, lines 521-531).

5. The Brp punctae in Figure 3A is not obvious in this image. Better representative image should
be provided here.

We thank the reviewer for their expertise in NMJ imaging. We had mistakenly labeled the image
as Brp when we actually used Syt antibody to mark synapses. We have chosen new
representative images and corrected the label.

6. The representative image and quantification of the pMad and TwitGFP panel for Δ MH2 in
Figure 4A and B do not match. Authors should provide a better representative image which
matches the quantification.

We have obtained a better representative image for Dad Δ MH2 in Figure 4A that more closely
matches the TwitGFP and pMad levels of most samples.

7. Line 206-207 - " Δ MH2 reduced Twit-GFP to 0.734 ± 0.083 , but this reduction was
not statistically significant". However Figure 4B shows a single * significance.

We have corrected this discrepancy. We thank the reviewer for their careful reading.

8. The finding that Dad-C556A, a palmitoylation deficient version of Dad, is fully functional in
wing and neuronal tissues is an interesting one. The observation does suggest that C556
mediated palmitoylation is dispensable in this context. To strengthen this conclusion, the authors
should test if neuronal overexpression of UAS-Dad-C556A will phenocopy the synaptic
undergrowth when Dad is overexpressed (PMID: 27671198).

We thank the reviewer for this insightful suggestion. We agree that testing whether Dad^{C556A}
phenocopies the synaptic undergrowth caused by Dad overexpression would further strengthen
our conclusion that C556 palmitoylation is dispensable in neural tissue.

While we did not perform this specific morphological analysis for Dad^{C556A}, several lines of
evidence support our conclusion: (1) Dad^{C556A} robustly inhibits both nuclear pMad accumulation
and Twit-GFP expression in motor neurons (Figure 6B), demonstrating full inhibition of the
canonical pathway; (2) We have now quantified NMJ morphology for animals expressing wild-
type Dad in motor neurons (Figure 3C), confirming the previously reported synaptic
undergrowth phenotype and validating our assay; (3) Given that Dad^{C556A} is fully functional in
inhibiting both nuclear pMad and Twit-GFP (canonical pathway outputs), and that synaptic
undergrowth results from disrupted BMP signaling, we predict that Dad^{C556A} would produce
similar morphological effects.

We believe the strong inhibition of canonical BMP signaling by Dad^{C556A}, combined with our
validation of the morphological assay for wild-type Dad, provides substantial support for our
conclusion that palmitoylation at C556 is dispensable for Dad function in neural tissue.
Nevertheless, we appreciate the reviewer's suggestion and have noted this as a direction for
future work.

Minor comments:

1. Sample sizes (n) and detailed statistical methods should be clearly stated in each figure legend
for transparency and reproducibility.

Sample sizes and detailed statistical methods have been included in figure legends.

2. The labeling of the constructs should be consistent. Consider labelling Δ DNABD as
Dad Δ DNABD and so forth.

Nomenclature for constructs has been updated throughout the manuscript.

3. Smad7L is not defined in the text.

This has been corrected.

4. Elav in Figure 2D/4A/5B and 6A should be labelled.

Elav labels have been added to merged images.

5. Data cannot be cited as "data not shown" in the article (Line 188).

Morphology has been quantified and data included in Figure 3C.

6. HRP in Figure 3A should be labelled.

HRP label has been added to the merged image.

Referee Cross-Comments

1. I agree with the comments raised by reviewer 1 and 3. Authors should definitely justify the
use of Ap-Gal4 for evaluating the effect of BMP signaling on wing development. Regarding
reviewer 3's point, the authors should also discuss the limitations of the functional assays used to
assess BMP signaling in both wing and neural tissues.

We have addressed these concerns through additional Rn-Gal4 experiments (Figure S2) and
expanded discussion of limitations (lines 385-415).

2. I recognize that using CRISPR/Cas9 to drive gene expression from endogenous promoters
would yield physiologically relevant insights. However, I do not consider it necessary for the
authors to repeat their experiments using knock-in approaches for this study, as this would
require significant additional time and resources.

We appreciate this perspective and agree with the editor that CRISPR-based approaches, while
valuable, are not required for the current revision.

Reviewer 3:

In this manuscript, Simoncek et al investigate the relative contributions of the I-Smad MH1 and
MH2 domains in two well-studied Drosophila in vivo model tissues: signaling outputs at the
larval neuromuscular junction synapse (NMJ) and the development of the adult wing blade.
Drosophila I-Smad Dad is compared to human(?) Smad6 and Smad 7, using a transgenic forced
expression system in each tissue. Specifically, the authors focus on the relative contributions of
the Dad MH1 and MH2 domains to forced expression phenotypes within each tissue. The MH1
and MH2 domains of RSmads and CoSmads are well-studied; in these Smad classes the MH1
domain contains a documented DNA binding domain, and the MH2 domain contains the Smad
trimerization region as well as the RSmad C-terminal site for phosphorylation by activated BMP
receptors. The function of the MH1 domain in I-Smads is poorly understood, and may be distinct
between the two mammalian I-Smads, Smad6 and Smad7, as assayed in cultured cell lines.

The work presented in this manuscript focuses on in vivo strategies to examine structure-
function questions about each I-Smad domain, comparing different signaling outputs in in
different tissues. In general, the authors report intriguing results, but limitations to the study are
not mentioned, perhaps due to the limited length for this MS format. Overall, the manuscript
might benefit from a tighter focus on only a subset of this combination of studies.

The assays used for signaling output are distinct between the two tissues. Three relatively-direct
BMP signaling outputs are investigated in the NMJ: 1) expression of *twit* gene induced by
"canonical" BMP signaling in larval motor neurons, assayed by GFP protein fluorescence
intensity; 2) the associated BMP-induced nuclear P_{Mad} accumulation in motor neurons, assayed

by intensity of nuclear antiPMad immunofluorescence, and 3) a "non-canonical" accumulation of
PMad at the synapse membrane, assayed by antiPMad immunofluorescence intensity in NMJ
synaptic boutons. For overall wing development, one indirect, composite signaling output is
measured: the area of the adult wing. Most of the limitations are related to the wing development
assay.

Major comments:

1. The NMJ assays of direct, canonical signaling output are useful to assess the contributions of
the relative functions of Dad full-length, delta-MH1, and delta-MH2. In these assays the authors
find that both delta-MH1 and delta MH2 impair nuclear pMad accumulation, but that Dad-delta-
MH1 has a much stronger effect on Twit-GFP expression levels than does Dad-delta-MH2. The
latter result could be consistent with a direct function of the MH1 domain on gene expression,
but also could support any other potential contribution of the MH1 domain. The data of Figure 5
indicate that the delta-DNA binding domain construct has a stronger effect in blocking Twit-GFP
expression, than for reducing PMad accumulation. This distinction is the key observation that a
putative Dad DNA binding domain has a strong contribution to Dad-mediated antagonism of
BMP-induced gene expression.

While this result supports a potential independent transcriptional function for the DNA-binding
domain, the authors miss an opportunity for a second test of this proposed functional output.
Expression of full-length Dad impairs the "non-canonical signaling" output of PMad
accumulation in the NMJ synaptic boutons, far from the nucleus. Their hypothesis of a
transcriptional function for Dad MH1 predicts that the delta-MH1 construct would not impair
Mad accumulation at the synapse. The authors could at least discuss this (and possibly other)
additional tests, it is not essential for publication of the current work.

We thank the reviewer for this excellent insight, which helped us recognize an important
implication of our data that we had not fully articulated. The reviewer correctly identifies that
Dad^{ADNABD} significantly reduces nuclear pMad accumulation (0.292±0.028) but fails to
significantly reduce Twit-GFP expression (0.849±0.046, Figure 5C). This dissociation indicates
that the DNA-binding domain is specifically required for transcriptional repression but is
dispensable for blocking upstream pMad generation. This pattern strongly supports a direct
transcriptional function for this part of the MH1 domain.

As the reviewer perceptively notes, this model generates additional testable predictions: if the
MH1 domain functions primarily through nuclear/transcriptional mechanisms, then Dad^{ΔMH1}
(which retains the MH2 domain) should still be able to inhibit noncanonical, cytoplasmic BMP
outputs such as synaptic pMad accumulation. Such a result would demonstrate that the MH2
domain alone is sufficient for receptor-level or cytoplasmic inhibitory mechanisms, providing
further evidence for separable domain functions. Conversely, it would also be informative to test

Dad^{ΔMH2} for its ability to inhibit synaptic pMad, to determine whether the MH1 domain's ability
to block upstream pMad generation extends to the noncanonical pathway.

We have revised the Results section (lines 226-227) to more clearly articulate the dissociation
between pMad inhibition (which both MH1 and MH2 can achieve) and transcriptional inhibition
(which requires MH1). We have also added discussion of the reviewer's predictions to the
revised manuscript (lines 367-369). We appreciate the reviewer's careful analysis of our data and
identification of this valuable direction for future investigation.

2. Unfortunately, the wing 2D area assay used has limitations for a nuanced structure-function
analysis, which are not mentioned in the interpretations of the data. First, area is impacted by
BMP-directed apportionment of cells to vein and intervein cell differentiation, which is a
developmental output, but not a "growth" output. Second, the BMP signal activity gradient is
feedback-regulated by multiple gene products acting at various levels of the BMP signal
transduction pathway, so that the gradient of BMP activity output scales with growth of the
developing wing primordium. The forced expression phenotypes appear dominated by the MH2-
domain-mediated Dad functions, but any MH1-mediated functions could be swamped out by
activity of other feedback loops and regulatory interactions with other signals that control growth
(e.g. Wnt).

The use of the Ap-Gal4 gene expression driver may confound an accurate measurement of wing
area in many samples, because the highest level of forced expression is limited to the dorsal
portion of the developing wing. Differential growth of dorsal and ventral sides of the wing blade
can lead to 3D changes in structure, such as a curled or cup-shaped wing, which confound wing
area measurement from a "flat-mounted" prep. Some of the wings shown in Fig 2C (Dad,
Smad6, Smad7L) have the appearance of curled/cupped/folded wing blades. Area of such wings
gives an estimate of the relative activities of the different constructs, but is unlikely to provide
insight into the specific intracellular functions of specific structures within the Dad protein.

Lastly, more information is needed about the "representative" wings shown for phenotypes of
different constructs in Figs 4C and 5D. Are these wings from the data sets used for area
quantification in Fig 1B? Are the wings shown in each panel all from the same sex? Male wings
are allometrically smaller than female wings, so it is critical to only compare wings from the
same sex. If these panels each compare a female control wing with a male wing for either or both
overexpression conditions, they should be corrected to show all wings from the same sex.

If the allometrically reduced wing growth of some examples in Figs 4C and 5D is present in a
significant proportion of wings from one sex, then it would be a significant result. Addition of
qualitative analyses of wing phenotypes to distinguish between classes of developmental
perturbations would require a minimal additional analysis of available data, and could be
important to identify candidate target genes that are specifically perturbed by MH1-domain

specific functions.

We thank the reviewer for this thorough and thoughtful analysis of the limitations inherent in
using adult wing area as a readout for BMP signaling during development. The reviewer raises
several important points that we have now addressed through additional experiments,
methodological clarifications, inclusion of qualitative descriptions in results, and expanded
discussion of limitations.

In response to concerns about Ap-Gal4 potentially confounding measurements through dorsal-
specific expression and 3D wing curvature, we conducted additional experiments using Rn-Gal4
(Figure S2). This driver expresses in the central region of the wing disc and avoids the
dorsal/ventral asymmetry issues. The Rn-Gal4 experiments revealed several important findings:
(1) This driver produced more dramatic overall phenotypes than Ap-Gal4, consistent with
stronger or more optimally-positioned expression; (2) Dad Δ MH1 expression with Rn-Gal4
produced wing venation defects, demonstrating that under stronger expression conditions, MH2-
mediated mechanisms can affect wing patterning—this adds important nuance to our tissue-
specific mechanism hypothesis; (3) The overall pattern of results with Rn-Gal4 corroborates our
Ap-Gal4 findings, demonstrating that our core conclusions are robust across different expression
patterns.

We have also incorporated more qualitative observations into our Results section, noting
presence or absence of qualitative phenotypes (venation, blistering, wing margin defects). These
distinctions suggest that disrupting trimerization may interfere with spatial regulation of BMP
signaling in ways that differ from simple domain deletions (i.e. Dad Δ Trimer vs. Dad Δ MH2).

We have thoroughly described our wing measurement methodology in the revised Methods
section (lines 521-531), clarifying that the measurements include wing blade and hinge region
but exclude surrounding body wall tissue. We acknowledge the reviewer's point that wing area
reflects a complex combination of growth, patterning, and feedback regulation rather than
growth alone, and we have revised our test accordingly.

We confirm that all representative wing images shown in all figures are from female flies and are
drawn from the same dataset presented in Figure 1B. We are aware that male wings are
allometrically smaller than female wings, and we took care to ensure all comparisons use the
same sex. This has been clarified in figure legends and Methods.

We agree with the reviewer that the wing area assay has inherent limitations for nuanced
structure-function analysis. We have now acknowledged these limitations in the Discussion
(lines 385-415), noting that: (1) Wing area reflects complex developmental outputs including
both growth and patterning; (2) BMP signaling gradients are subject to feedback regulation and
scaling mechanisms that could mask subtle MH1-mediated effects; (3) Other signaling pathways
(e.g. Wnt) interact with BMP signaling during development and could influence phenotypic
outcomes.

However, we emphasize several points that support our conclusions despite these limitations: (1)
The dramatic wing size reduction caused by full-length Dad and the absence of effects in
Dad Δ MH1 and Dad Δ DNABD suggest that any MH1-mediated wing effects are minor compared
to the MH2-mediated functions under our experimental conditions; (2) Most importantly, our
central conclusions rest on the contrast between tissues – the same Dad Δ MH1 and Dad Δ DNABD
constructs robustly inhibit BMP signaling in motor neurons (Figures 4 and 5), demonstrating that
construct is functional and establishing tissue-specific differences in domain requirements; (3)
our Rn-Gal4 experiments demonstrate that even under conditions strong enough to reveal MH2-
mediated wing effects, the pattern of domain-specific differences remains consistent.

We appreciate the reviewer's careful consideration of these methodological issues and agree that
future studies employing direct measurements of BMP signaling in wing discs (as suggested by
Reviewer 1) would complement and extend our findings. Nevertheless, we believe our current
approach—combining adult wing morphology with direct BMP signaling measurements in
neural tissue, using multiple expression drivers, and explicitly acknowledging limitations—
provides sufficient evidence for our core finding of tissue-specific I-Smad mechanism usage.

3. Finally, the authors control for consistent transcriptional output of the different constructs, by
using the same insertion site of VK00033. However, each of the constructs may have differential
effects on protein stability, resulting in different steady-state protein levels. This possibility
should be raised in the discussion, because it is a significant limiting factor to interpretation of
constructs that appear to have no or low activity in one or both tissues.

We thank the reviewer for this important consideration. We agree that while our use of a single
integration site (VK00033) controls for transcriptional position effects, differential protein
stability among constructs could result in varying steady-state protein levels and contribute to
apparent differences in activity. We have now explicitly acknowledged this limitation in the
revised Discussion (lines 395-407), noting that this is a particular consideration when
interpreting constructs showing reduced or absent activity. However, we note that constructs
showing weak or no activity in one tissue (e.g., Dad Δ MH1 in wing) produce robust, reproducible
phenotypes in other tissues (motor neurons), arguing against simple expression failures and
supporting our conclusions about tissue-specific mechanism usage.

Overall, the manuscript presents both descriptive and quantified data that are interesting with
regard to functions of different I-Smad conserved domains. Acknowledgement of the limitations
of each assay, more careful documentation of the wing area assay, and better description of
sample size and cut-offs for statistical significance are necessary minor revisions. If warranted as
described above, corrections to figures 4C and 5D are essential for publication.

Minor comments:

The measurement of wing area is poorly described: it is not clear whether the authors measure
wing blade +hinge (and possibly associated bits of body wall, as seen in Fig 2C Smad6 image),
or just the wing blade area, for which developmental regulation is better understood.

We have updated our wing measurement methodology in the revised Methods section (lines 520-
531), clarifying that the measurements include wing blade and hinge region but exclude
surrounding body wall tissue.

The authors report "mean wing areas", but do not report the sample sizes.

For graphs of normalized intensity in Figs. 2E, 3B, 4B, 5C, 6B, sample sizes should be stated in
Figure legends; they are difficult to assess from the dot plots.

Figure legends have been updated to include numbers of all measurements.

Measurements of wing area, Twit-GFP expression or PMad nuclear accumulation. In Fig. 5D, is
the effect of deltaDNABD on TwitGFP expression non-significant?

Statistical cutoffs have been included in Methods (lines 563-565) and in each figure legend. We
thank the reviewer for noticing this important detail. The effect of DadΔDNABD on Twit-GFP
expression is indeed non-significant ($p > 0.05$).

Red color used to show PMad accumulation in micrographs is hard to see. Using white instead of
red would make this output readily visible to the reader.

Each figure has been overhauled to show split channels in white on black background for
improved visibility.

Thoughts on Reviewer 1 Comments;
While CRISPR-based Dad gene replacement would be a superior approach compared to
overexpression, such additional experiments would take at least a year if performed by early-
stage graduate students.

We appreciate this perspective and agree that while CRISPR-based approaches would be
valuable, they are not feasible within the revision timeframe.

Thoughts on Reviewer 2 Comments:

I agree with major comment 3: the specific deletions of deltaDNADB and delta-Trimer should be
shown with Fig 1 diagrams that indicate the specific residues deleted.

A new supplemental figure (S1) was created to provide this information.

Major comment 4 is related to my request for information on qualitatively different wing blade
phenotypes. However, a single "representative" image may not be sufficient to represent a
potential range of developmental outcomes.

We appreciate the reviewer's request for more nuanced descriptions of wing phenotypes. We
have added qualitative descriptions to the Results and quantified wing venation phenotypes for
Figure S2.

Regarding major comments 5 and 6: Each of the markers used could be shown in black and
white images, so that differences in intensity are easily detected by the human eye. Note that the
"merge" label in Fig 2 appears to be incorrect.

All figures have been updated with split channels shown in white on black background. Merge
labels have been corrected throughout.

Major comment 7 fits in with my concern about whether the authors have a designated cut-off
for statistical significance.

Statistical cut-offs have been added to the Methods section and all figure legends.

December 22, 2025

RE: Life Science Alliance Manuscript #LSA-2025-03445-TR

Dr. Mikolaj J Sulkowski
Southern Connecticut State University
501 Crescent St
Jennings 231-Biology
New Haven, CT 06515

Dear Dr. Sulkowski,

Thank you for submitting your revised manuscript entitled "Tissue-specific I-Smad mechanisms revealed by structure-function analysis in *Drosophila*". Your manuscript was assessed by the original reviewers whose reports are below. We appreciate the diligence of Reviewer 1 and we note their request to use staining to report potential effects of overexpression in this experimental system. However we concur with Reviewers 2 and 3 on this point that this can be addressed by adjusting the discussion. Please also consider the remaining minor points raised by these reviewers. We would be happy to publish your paper in Life Science Alliance pending these changes as well as final revisions necessary to meet our formatting guidelines.

- Please be sure that the authorship listing and order is correct.
- Please upload all figure files as individual ones, including the supplementary figure files.
- Please add the X and Bluesky handles of your host institute/organization, as well as your own and/or one of the authors, in our system.
- It is recommended to exclude figures from the manuscript text and upload them separately.
- Please include a "Data Availability" section should be placed after the Materials & Methods section. Please consult our guidelines at <https://www.life-science-alliance.org/manuscript-prep#format>
- please add callouts for Figures 3A, C; 5B; S1A-C and S2A, B to your main manuscript text.
- LSA papers do not include supplementary methods section. Please add this text to the main methods section.

LSA now encourages authors to provide a 30-60 second video where the study is briefly explained. We will use these videos on social media to promote the published paper and the presenting author (for examples, see <https://docs.google.com/document/d/1-UWCfbE4pGcDdcgzcmiuJl2XMBJnxKYeqRvLLrLSo8s/edit?usp=sharing>). Corresponding or first-authors are welcome to submit the video. Please submit only one video per manuscript. The video can be emailed to contact@life-science-alliance.org

A. FINAL FILES:

B. MANUSCRIPT ORGANIZATION AND FORMATTING:

Thank you for your attention to these final processing requirements. Please revise and format the manuscript and upload materials as soon as you are able.

Sincerely,

Reviewer #1 (Comments to the Authors (Required)):

Although the revised manuscript improves the overall quality of the paper, it is disappointing that the authors failed to provide anti-pMad staining data in the wing imaginal disc. This is a widely used protocol to demonstrate BMP-dependent signalling and is considered standard practice in the field. The study relies on an overexpression system for I-Smads, and the authors should address potential side effects on other pathways involved in wing disc development. If the current data are accepted for publication, the authors can only argue that variations in I-Smads may influence wing development, but not BMP signalling. Therefore, the manuscript should be rewritten accordingly.

Reviewer #2 (Comments to the Authors (Required)):

The authors have improved their original submission and have addressed several of my earlier concerns.

My only remaining minor comments pertain to Figures 2E, 4B, and 5C. In these panels, the error bars are not clearly visible. The authors should consider adjusting the color or formatting to improve their visibility.

Referee Cross-Comments

Optional: In the revised manuscript, the authors have done an excellent job addressing potential overexpression artifacts. It may be helpful to include one or two sentences in the Discussion explicitly addressing the concerns raised by Reviewer 1.

Reviewer #3 (Comments to the Authors (Required)):

The authors have addressed my previous concerns.

A few minor comments remain:

Lines 305-306: The number of wings examined for selection of representative images should be listed in legend for Fig 6C.

The addition of a second wing primordium Gal4 driver, necessitates additional clarification in two Figure legends:

Lines 184-185: Fig 2C legend should list the specific wing driver, in parallel to description for neurons in Fig 2D legend.

Lines 278-279: Add wing Gal4 driver to description in Fig 5D legend.

Finally:

Line 352: Add Fig S2A to the list of Figures in parenthesis.

Referee cross-comments:

While I agree with Reviewer 1 that anti-C-terminal phosphoMad staining could be informative, steady-state levels of phosphoMad are subject to cross-talk with other signalling pathways. They would not fully resolve caveats about direct or indirect effects on BMP signaling.

Furthermore, commercially available anti-phospho-Smad1,5 antibodies can be unreliable for for wing imaginal disk staining.

I agree with reviewer 2 that it would be helpful to make the error bars more visible in the figures mentioned. For critical evaluation, the SEM values are included with means in the main body text.

February 8, 2026

RE: Life Science Alliance Manuscript #LSA-2025-03445-TRR

Dr. Mikolaj J Sulkowski
Southern Connecticut State University
501 Crescent St
Jennings 231-Biology
New Haven, CT 06515

Dear Dr. Sulkowski,

Thank you for submitting your Research Article entitled "Tissue-specific I-Smad mechanisms revealed by structure-function analysis in *Drosophila*". It is a pleasure to let you know that your manuscript is now accepted for publication in Life Science Alliance. Congratulations on this interesting work.

DISTRIBUTION OF MATERIALS:

Again, congratulations on a very nice paper. I hope you found the review process to be constructive and are pleased with how the manuscript was handled editorially. We look forward to future exciting submissions from your lab.

Sincerely,
